# Anisotropic higher rank $\mathbb{Z}_N$ topological phases on graphs

**Hiromi Ebisu[1] and Bo Han[2]**

**1** Department of Physics and Astronomy, Rutgers, The State University of New Jersey,
School of Arts and Sciences, New Jersey, USA
**2** Department of condensed matter Physics, Weizmann Institute of Science, Rehovot, Israel

## Abstract

We study unusual gapped topological phases where they admit $\mathbb{Z}_N$ fractional excitations in the same manner as topologically ordered phases, yet their ground state degeneracy depends on the local geometry of the system. Placing such phases on 2D lattice, composed of an arbitrary connected graph and 1D line, we find that the fusion rules of quasiparticle excitations are described by the Laplacian of the graph and that the number of superselection sectors is related to the kernel of the Laplacian. Based on this analysis, we further show that the ground state degeneracy is given by $\left[ N \times \prod_i \gcd(N, p_i) \right]^2$, where $p_i$'s are invariant factors of the Laplacian that are greater than one and gcd stands for the greatest common divisor. We also discuss braiding statistics between quasiparticle excitations.

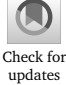

# 1 Introduction

Topologically ordered phases are novel phases beyond the paradigm of the standard Landau-Ginzburg theory [1–4] and have been one of the central topics in condensed matter physics for decades. There are many salient features in these phases, like deconfined fractionalized excitations (anyons) [2,5,6], and they may find many ramifications in different fields of physics, such as quantum information [7,8] and high energy physics [9–11].

While a plethora of progress has been made towards complete understanding topologically ordered phases from both of theoretical and experimental point of view, recently, new types of topologically ordered phases have been proposed, which are often called fracton topological phases in the literature [12–14]. One of the intriguing properties of the fracton topological phases is that ground state degeneracy (GSD) depends on the UV lattice spacing, which is contrasted with conventional topologically ordered phases where GSD depends only on the global topology of the manifold. This property can be intuitively understood by that fractional quasiparticle excitations are affected by local geometry of the system. In other words, the local geometry imposes a mobility constraint on the fractional excitations, giving rise to sub-extensive dependence of the GSD. In this view, fracton topological phases may open a possibility to explore new geometric phases.

In this work, we consider unusual 2D gapped $\mathbb{Z}_N$ topological phases with a distinct feature that while they admit $\mathbb{Z}_N$ fractional excitations in the same way as the topologically ordered phases, their GSD depends on the local geometry of the system, similar to the fracton topological phases. We emphasize that our model is different from fracton topological phases as it does not show the sub-extensive GSD dependence. Rather, it exhibits unusual GSD dependence on $N$ and geometry of the lattice. To investigate interplay between fractional excitations and geometry of the system, we place such phases on 2D lattice consisting of arbitrary connected graph, a pair consisting of a set of vertices and a multiset of edges, and 1D line. A systematic study on geometric aspects of the fractional excitations can be accomplished by resorting to the well-developed algebraic tools of graph theory, such as the Laplacian and Picard group [15,16]. (see also Ref. [17] for a study on 3D fracton topological phases on the Cayley trees and Ref. [18] for analysis on the Lifshitz theory on a graph and complexity.)

We find that fusion rules of fractional excitations are described by the Laplacian of the graph and that the number of superselection sectors (i.e. the number of distinct types of fractional excitations) is associated to kernel of the Laplacian. Further analysis shows that GSD depends on the greatest common divisor of $N$ and invariant factors of the Laplacian.

Laplacian plays an important role in graph theory. For instance, one can study connectivity of a graph by evaluating its eigenvalues [19,20]. In our context, the Laplacian is crucial to characterize fusion rules of fractional charges and GSD dependence. We also study braiding statistics between electric and magnetic charges and find that it is described by matrices with which the Laplacian is transformed into a diagonal form, known as the Smith normal form. These results are summarized in (26) and (40).

The outline of this paper is as follows. In Sec. 2, after reviewing the notations and formulations in graph theory, we introduce a 2D lattice, which is a product of an arbitrary connected graph and 1D lattice, and model Hamiltonian. In Sec. 3, we discuss behaviours of fractional excitations of our model. We derive the fusion rules of the excitations, and the GSD dependence on the graph, based on the formulation of the Laplacian. In Sec. 4, we demonstrate several examples of the graph (cycle graph and complete graph) to see more transparently how our results presented in the previous section works. Finally, in Sec. 5, we conclude our work with a few remarks for future directions.

## 2 Model

In this section, we introduce our lattice and model. Since these are described by graph and the Laplacian, we briefly go over notations and formulations of graph theory, especially the properties of the Laplacian, before introducing the lattice and model Hamiltonian. Our model shares the same features as the $\mathbb{Z}_N$ toric code [8], e.g., Hamiltonian consists of mutually commuting terms and there are two types of excitations carrying fractional charges. However, our model has the distinct property from the toric code: depending on $N$ and graph, excitations are subject to a mobility constraint in the $x$-direction, giving rise to unusual GSD.

### 2.1 Notations of graph and Laplacian

Let us first introduce a graph $G = (V, E)$ which is a pair consisting of a set of vertices $V$ and a set of edges $E$ comprised of pairs of vertices $\{v_i, v_j\}$. Throughout this work, we assume that the graph is *connected*, meaning there is a path from a vertex to any other vertex, and that the graph does not have an edge that emanates from and terminates at the same vertex. We also introduce two quantities, $\deg(v_i)$ and $l_{ij}$, which play pivotal roles in this paper. The former one, $\deg(v_i)$ denotes *degree* of the vertex $v_i$, i.e. the number of edges emanating from the vertex $v_i$ and the latter one, $l_{ij}$ represents the number of edges between two vertices $v_i$ and $v_j$ (We have $l_{ij} = 0$ when there is no edge between two vertices, $v_i$ and $v_j$.). Using these two quantities, *Laplacian matrix* of the graph, which is the analogue of the second order derivative operator $\partial_x^2$ on a graph, is defined. For a given graph $G = (V, E)$, the Laplacian matrix $L$ (which we abbreviate as Laplacian in the rest of this work) is the matrix with rows and columns indexed by the elements of vertices $\{v_i\} \in V$, with

$$L_{ij} = \begin{cases} \deg(v_i), & (i = j), \\ -l_{ij}, & (i \neq j). \end{cases} \tag{1}$$

The Laplacian is singular due to the connectivity of the graph. (Summing over all rows or columns gives zero.) As an example, the Laplacian of the cycle graph $C_3$ (i.e. a triangle) consisting of three vertices and three edges, where there is a single edge between a pair of vertices, is given by

$$L = \begin{pmatrix} 2 & -1 & -1 \\ -1 & 2 & -1 \\ -1 & -1 & 2 \end{pmatrix}.$$

It is known that by introducing invertible matrices over integer $P, Q$ corresponding to linear operations on rows and columns of the Laplacian, respectively, the Laplacian of any connected graph can be transformed into a diagonal form (*Smith normal form*):

$$PLQ = \mathrm{diag}(u_1, u_2, \cdots, u_{n-1}, 0) := D, \tag{2}$$

where $u_i$ represents positive integers, satisfying $u_i | u_{i+1}$ for all $i$ (i.e. $u_i$ divides $u_{i+1}$ for all $i$) [16]. Since the Laplacian is singular, the last diagonal entry is zero. The diagonal element $u_i$ referred to as the *invariant factors* of the Laplacian, plays a pivotal role in the graph theory. For instance, their product is equivalent to the number of spanning trees, which are connected subgraphs where there is a unique path from a vertex to any other vertex [15]. In our work, $u_i$ is crucial quantity to characterize the superselection sectors of fractional excitations.



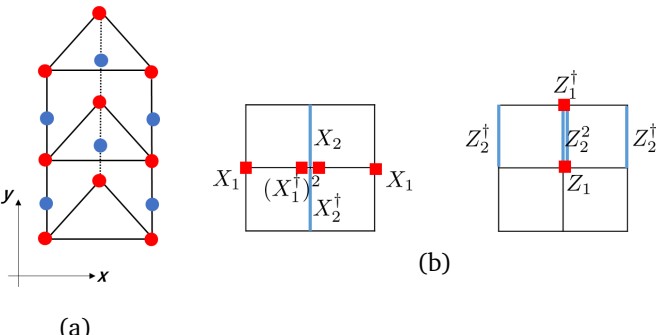

Figure 1: (a) One example of the 2D lattice, consisting of cycle graph ($C_3$) in the $x$-direction and 1D lattice in the $y$-direction. (b) Two types of terms introduced in (4), $V_{(v_i,y)}$ (left) and $P_{(v_i,y+1/2)}$ (right) in the case of the square lattice.

## 2.2 Lattice and Hamiltonian

The 2D lattice is defined by a product of a graph that runs across in the $x$-direction and 1D lattice going in the $y$-direction.[1] More explicitly, for a given graph, the 2D lattice is constructed by stacking the copies of the graph and add vertical edges between the adjacent graph. As an example, we demonstrate the case where the graph is cycle graph $C_3$ in Fig. 1a. We place two types of quantum states, each of which takes discrete $N$ ($\geq 2$) values, i.e. generalized qubits ($\mathbb{Z}_N$ clock states) on this lattice. The first clock states are located at vertices of the graph (red dots in Fig. 1a) whereas the second ones are at vertical edges (blue dots in Fig. 1a). We denote the coordinate of the first clock states by $(v_i, y)$ where $v_i$ represents a vertex of the graph and $y$ does the height taking integer values in the unit of lattice spacing. Analogously, the coordinate of the second clock states are denoted by $(v_i, y + \frac{1}{2})$, where the second element corresponds to the edge between vertices located at $(v_i, y)$ and $(v_i, y + 1)$.

Having defined the 2D lattice, we introduce the Hamiltonian. We represent basis of the two types of the clock states as $|\omega\rangle_1$ and $|\omega\rangle_2$ with $\omega$ being $N$-th root of unity, i.e, $\omega = e^{2\pi i/N}$, and $\mathbb{Z}_N$ shift and clock operators (they become Pauli operators when $N = 2$) of the first and second clock states as $\{Z_i, X_i\}$ ($i = 1, 2$). Here we have introduced subscript $i = 1, 2$ to distinguish operators that act on the clock clock states at vertices and the ones at vertical edges. These operators satisfy the following relation ($I_i$ denotes the identity operator)

$$X_i^N = Z_i^N = I_i, \qquad Z_i |\omega\rangle_i = \omega |\omega\rangle_i, \qquad X_i Z_j = \omega Z_j X_i \delta_{i,j}. \tag{3}$$

With these notations, we define following two types of operators at each vertex and edge

$$V_{(v_i,y)} := X_{2,(v_i,y+1/2)} X_{2,(v_i,y-1/2)}^\dagger (X_{1,(v_i,y)}^\dagger)^{\deg(v_i)} \prod_j (X_{1,(v_j,y)})^{l_{ij}},$$

$$P_{(v_i,y+1/2)} := Z_{1,(v_i,y+1)}^\dagger Z_{1,(v_i,y)} Z_{2,(v_i,y+1/2)}^{\deg(v_i)} \prod_j (Z_{2,(v_j,y+1/2)}^\dagger)^{l_{ij}}. \tag{4}$$

Here, $\deg(v_i)$ is the number of edges in the $x$-direction which emanate from the vertex $v_i$ whereas $l_{ij}$ gives the number of edges in the $x$-direction between two vertices with coordinate $(v_i, y)$ and $(v_j, y)$. We demonstrate examples of these two types of terms in Fig. 1b in the case of the square lattice, where at any vertex $v_i$, (without taking into account boundary) we

---

[1] We regard the graph as 1D, since it consists of 1- and 0-simplices (which correspond to edges and vertices, respectively).

have $\deg(v_i) = 2$ and $l_{ij} = 1$ $(j = i \pm 1), l_{ij} = 0$ (else). Hamiltonian is defined by

$$H = -\sum_{(v_i,y)} V_{(v_i,y)} - \sum_{(v_i,y+1/2)} P_{(v_i,y+1/2)} + h.c. \tag{5}$$

Note that Hamiltonian is described by $\mathbb{Z}_N$ shift and clock operators and the two types of quantities, $\deg(v_i)$ and $l_{ij}$. Especially, the latter ones also enter in the Laplacian of the graph (1), allowing us to systematically investigate physical properties of Hamiltonian by resorting to formulations of the Laplacian.

Each term in Hamiltonian (5) commutes with one another. To verify this, for a given coordinate $(v_i, y)$, one has to check commutation relation between $V_{(v_i,y)}$ and some of the second terms in (5) which has overlapping support with the one of $V_{(v_i,y)}$. Such terms are given by $P_{(v_i,y+1/2)}$, $P_{(v_i,y-1/2)}$, $P_{(v_j,y+1/2)}$, $P_{(v_j,y-1/2)}$, where $v_j$ represents a vertex adjacent to $v_i$. Using (3) and (4), we have

$$V_{(v_i,y)}P_{(v_i,y+1/2)} = \omega^{\deg(v_i)}\omega^{-\deg(v_i)}P_{(v_i,y+1/2)}V_{(v_i,y)} = P_{(v_i,y+1/2)}V_{(v_i,y)},$$
$$V_{(v_i,y)}P_{(v_j,y+1/2)} = \omega^{l_{ij}}\omega^{-l_{ij}}P_{(v_j,y+1/2)}V_{(v_i,y)} = P_{(v_j,y+1/2)}V_{(v_i,y)}.$$

Similarly, one can check other commutation relations and find that every term in (5) indeed commutes. The ground state of this Hamiltonian (5), $|\Omega\rangle$ satisfies

$$V_{(v_i,y)}|\Omega\rangle = P_{(v_i,y+1/2)}|\Omega\rangle = |\Omega\rangle, \quad \forall\, V_{(v_i,y)}, P_{(v_i,y+1/2)},$$

that is, the ground state satisfies $V_{(v_i,y)} = 1$ and $P_{(v_i,y+1/2)} = 1$ at any coordinate.

In what follows, we discuss quasiparticle excitations of this model. As we will see, the excitations show unusual behavior in $x$-direction compared with conventional topologically ordered phases, giving rise to novel GSD dependence on the lattice. In the next subsection, we start with the simplest case by setting $N = 2$ and square lattice to intuitively understand this feature and in the later sections, we present more detailed discussions on the excitations based on formalism of graph theory.

## 2.3  Simplest case: $N = 2$ and the square lattice

To extract more intuition from Hamiltonian (5), we consider the case with $N = 2$ and the square lattice, which corresponds to setting the graph to be the cycle graph $C_n$. This graph consists of $n$ vertices placed in a cyclic order so that adjacent vertices are connected by a single edge. By setting $N = 2$ in (3), the terms (4) become simplified:

$$V_{(v_i,y)} = X_{2,(v_i,y+1/2)}X_{2,(v_i,y-1/2)}X_{1,(v_{i-1},y)}X_{1,(v_{i+1},y)},$$
$$P_{(v_i,y+1/2)} = Z_{1,(v_i,y+1)}Z_{1,(v_i,y)}Z_{2,(v_{i-1},y+1/2)}Z_{2,(v_{i+1},y+1/2)}, \quad (1 \le i \le n). \tag{6}$$

Here the vertices $\{v_i\}$ $(1 \le i \le n)$ are placed in cyclic order and we conventionally set $v_{n+1} = v_1$ and $v_{-1} = v_n$. We portray these terms in Fig. 2a. Hamiltonian (5) is defined by using these terms. It is straightforward to check each term in Hamiltonian commutes. The ground state satisfies $V_{(v_i,y)} = 1$ and $P_{(v_i,y+1/2)} = 1$ at any coordinate. From (6) and Fig. 2a, one can easily associate mutual commuting terms in Hamiltonian to the ones of the $\mathbb{Z}_2$ toric code [8] except the fact that in the $x$-direction, the next nearest neighboring Pauli operators $X_1$ or $Z_2$ enter in the terms (6).

With this difference in mind, let us look at the excitations of Hamiltonian. Analogous to the $\mathbb{Z}_2$ toric code, one can act $X_{1(2)}$ and $Z_{1(2)}$ operators to create an excitation carrying "electric" and "magnetic" charge, respectively. Consider acting a single $Z_1$ operator at $v_i$ on the ground state, i.e, applying $Z_{1,(v_i,y)}$ on the ground state. It violates $V_{1,(v_{i-1},y)} = 1$ and $V_{1,(v_{i+1},y)} = 1$,

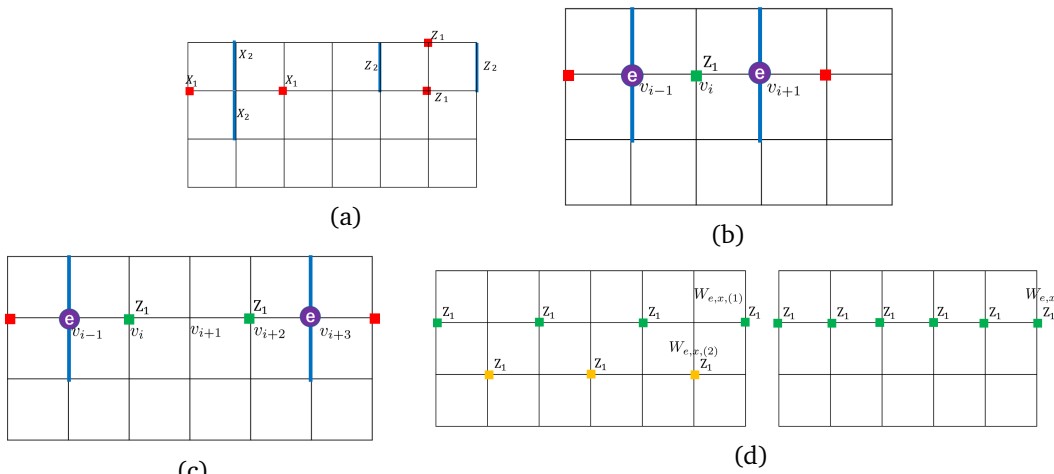

Figure 2: Configurations of terms which constitute of Hamiltonian and the form of excitations in the case of $N = 2$ and the square lattice. (a) Two terms given in (6). (b) When applying $Z_{1,(v_i,y)}$ (green square), $V_{(v_{i\pm1},y)} = 1$ is violated, creating a pair of electric charges (purple dots). (c) Applying further $Z_{1,(v_{i+1},y)}$ in (b), the trajectory of electric charges is stretched. (d) Left: when $n$ is even ($n = 6$ in this figure), there are two distinct closed loops of electrics charges (green and yellow squares). Right: when $n$ is odd ($n = 5$ in this figure), there is only one closed loop (green squares). In both of left and right figures, the rightmost vertical edges and vertices are identified with the leftmost ones.

giving rise to a pair of electric charges as demonstrated in Fig. 2b. If we further act another $Z_1$ operator at $v_{i+2}$, the trajectory of the electric charges is stretched so that one of them jumps between next nearest vertices (Fig. 2c). Note that such an unusual behavior of the electric charge can be seen only in the $x$-direction. As for the $y$-direction, the behavior of electric charge is the same as the toric code: the trajectory of the electric charge is formed in such a way that the electric charge hops between nearest edges in the $y$-direction. The behavior of magnetic charges can be similarly discussed.

One can easily evaluate the GSD in this case, imposing the periodic boundary condition in the $y$-direction. Similarly to the toric code, one can successively act $Z_1$ operators on the ground state so that a non-contractible closed loop of the electric charge is formed, which is responsible for the non-trivial GSD. In the case of $n$ being even, due to the fact that the electric charge hops between next nearest vertices, there are two closed loops of the electric charge in the $x$-direction, the one formed by string of $Z_1$ operators at even vertices and the other at odd ones, denoted by $W_{e,x,(1)}$ and $W_{e,x,(2)}$, respectively. On the contrary, in the case of $n$ being odd, there is only one closed loop, $W_{e,x}$. the configurations of these loops are portrayed in Fig. 2d. One can count analogously the number of closed loops of the magnetic charge, and obtain the same result: in the case of $n$ even, there are two distinct loops, $W_{m,x,(1)}$ and $W_{m,x,(2)}$ whereas in the case of $n$ odd, there is only one, $W_{m,x}$. As we mentioned previously, the behavior of excitations in the $y$-direction is the same as the $\mathbb{Z}_2$ toric code, hence, these closed loops that we consider are deformable to the ones shifted above or below in the $y$-direction by applying sets of terms given in (6).

In the torus geometry, the GSD is equivalent to the number of topological excitations, i.e. the number of superselection sectors which are distinguishable by closed loops of electric and magnetic charges [8]. Based on this fact, in the case of $n$ being even, there are four closed loops of $\mathbb{Z}_2$ charge, $W_{e,x,(1)}$, $W_{e,x,(2)}$, $W_{m,x,(1)}$, and $W_{m,x,(2)}$, thus the GSD is given by $4^2 = 16$.

On the other hand, in the case of $n$ being odd, there are two closed loops of $\mathbb{Z}_2$ charge, $W_{e,x}$ and $W_{m,x}$, therefore we find that the GSD is given by $2^2 = 4$. Summarizing,

$$
\text{GSD} = \begin{cases} 16, & (n \text{ even}), \\ 4, & (n \text{ odd}). \end{cases}
$$

This result is consistent with a formula (26) that we are going to prove in the next section.

## 3 Ground state degeneracy and quasiparticle statistics

In this section, we study distinct quasiparticle excitations of the model. To this end, we show that the fusion rules of the quasiparticle excitations are succinctly described by the Laplacian (1). Using this result, we also show that the number of superselection sectors and the GSD of our model depends on $N$ and the great common divisor of $N$ and invariant factors of the Laplacian.

### 3.1 Fusion rules and Laplacian

In this subsection, we study excitations of our model by acting a single $X_{1(2)}$ or $Z_{1(2)}$ operator on the ground state. There are two types of fractional excitations in our model, referred to as $\mathbb{Z}_N$ "electric" and "magnetic" charges that violate $V_{(v_i,y)} = 1$ and $P_{(v_i,y+1/2)} = 1$ defined in (4), respectively. We label these two excitations at coordinate $(v_i, y)$ and $(v_i, y + 1/2)$, whose eigenvalue of $V_{(v_i,y)}$ and $P_{(v_i,y+1/2)}$ is $\omega$, by $e_{(v_i,y)}$ and $m_{(v_i,y+1/2)}$. Also, we label their conjugate with eigenvalue $\omega^{-1}$ by $\overline{e}_{(v_i,y)}$ and $\overline{m}_{(v_i,y+1/2)}$. We interchangeably use the notation $e_{(v_i,y)}^{-1} = \overline{e}_{(v_i,y)}$, $m_{(v_i,y+1/2)}^{-1} = \overline{m}_{(v_i,y+1/2)}$.[2]

We first concentrate on the electric charges. Suppose we introduce an excited state by acting an operator $Z_{1,(v_i,y)}$ at $(v_i, y)$ on the ground state. From (4), such a state violates $V_{(v_i,y)} = 1$ and $V_{(v_j,y)} = 1$ with $v_j$ being adjacent vertices to $v_i$, giving eigenvalue $\omega^{-\deg(v_i)}$ and $\omega^{l_{ij}}$, respectively. More precisely, we have

$$
V_{(v_i,y)}(Z_{1,(v_i,y)}|\Omega\rangle) = \omega^{-\deg(v_i)}(Z_{1,(v_i,y)}|\Omega\rangle), \quad V_{(v_j,y)}(Z_{1,(v_i,y)}|\Omega\rangle) = \omega^{l_{ij}}(Z_{1,(v_i,y)}|\Omega\rangle),
$$

where $|\Omega\rangle$ denotes the ground state. Hence, by acting $Z_{1,(v_i,y)}$ on the ground state, we schematically obtain the fusion rule of the electric charges:

$$
I \to (\overline{e}_{(v_i,y)})^{\deg(v_i)} \otimes \prod_j (e_{(v_j,y)})^{l_{ij}}, \tag{7}
$$

where $I$ denotes vacuum sector. The fusion rule (7) is the generalization of the one studied in conventional topologically ordered phases, where a pair of quasiparticle excitations are created.

To discuss more systematically the fusion rules in our model, it is useful to introduce the Laplacian of the graph given in (1). For a graph $G(V, E)$ at given $y$, we define $n$-dimensional vector where each entry takes $\mathbb{Z}_N$ value by

$$
\mathbf{r} = (r_1, r_2, \cdots, r_n)^T \in \mathbb{Z}_N^n, \tag{8}
$$

---

[2]Throughout this paper, we focus on excitations of the first two terms given in Hamiltonian (5). This is because the excitations of the rest terms are conjugate of the ones of the first two terms.

with $n$ being the number of vertices, from which we introduce multiple sets of $Z_1$ operators, $Z_{1,(v_1,y)}^{r_1} Z_{1,(v_2,y)}^{r_2} \cdots Z_{1,(v_n,y)}^{r_n}$ acting on the ground state. Also, introducing fundamental basis of vectors $\{\lambda_i\}$ as $\lambda_i = (\underbrace{0,\cdots,0}_{i\text{-}1}, 1, \underbrace{0,\cdots,0}_{n-i})^T \in \mathbf{r}$, the fusion rule (7) is rewritten as

$$I \to e_{(v_1,y)}^{a_1} \otimes e_{(v_2,y)}^{a_2} \otimes \cdots \otimes e_{(v_n,y)}^{a_n} \, (a_i \in \mathbb{Z}_N), \tag{9}$$

with

$$\mathbf{f}_e := (a_1, a_2, \cdots, a_n)^T = -L\lambda_i. \tag{10}$$

Note that in the fusion rule (9), charge conservation is satisfied, i.e, $\sum_i a_i = 0 \, (\mathrm{mod} N)$ as the Laplacian is singular (summing over matrix elements along $i$-th column gives zero).

We can similarly discuss the fusion rules of the electric charges induced by applying multiple sets of $Z_1$ operators on the ground state. When we apply $Z_{1,(v_1,y)}^{r_1} Z_{1,(v_2,y)}^{r_2} \cdots Z_{1,(v_n,y)}^{r_n}$ on the ground state, characterized by vector $\mathbf{r}$ (8), the fusion rule of the electric charges has the same form as (9) by setting

$$\mathbf{f}_e = -L\mathbf{r}. \tag{11}$$

So far we have considered fusion rules of the electric charges in $x$-direction. As for the fusion rules in the $y$-direction, by acting $Z_{2,(v_i,y+1/2)}$ on the ground state, a pair of electric charges are created, giving

$$I \to \overline{e}_{(v_i,y+1)} \otimes e_{(v_i,y)}, \tag{12}$$

sharing the same fusion rule as the one in the toric code.

By the similar line of thoughts, we obtain the fusion rules of magnetic charges in the $x$-direction. Introducing a vector $\mathbf{s} = (s_1, s_2, \cdots, s_n)^T \in \mathbb{Z}_N^n$, corresponding to an excited state by applying $X_{2,(v_1,y+1/2)}^{s_1} X_{2,(v_2,y+1/2)}^{s_2} \cdots X_{2,(v_n,y+1/2)}^{s_n}$ on the ground state, one can associate an excited state induced by applying a single $X_2$ operator, $X_{2,(v_i,y+1/2)}$ to a fundamental basis of vector, $\eta_i = (\underbrace{0,\cdots,0}_{i\text{-}1}, 1, \underbrace{0,\cdots,0}_{n-i})^T \in \mathbf{s}$. The fusion rule of the magnetic charges reads

$$I \to m_{(v_1,y+1/2)}^{b_1} \otimes m_{(v_2,y+1/2)}^{b_2} \otimes \cdots \otimes m_{(v_n,y+1/2)}^{b_n}, \quad (b_i \in \mathbb{Z}_N), \tag{13}$$

with

$$\mathbf{f}_m := (b_1, b_2, \cdots, b_n)^T = L\eta_\mathbf{i}. $$

In the $y$-direction, a pair of magnetic charges are created by applying $X_{1,(v_i,y)}$, yielding

$$I \to \overline{m}_{(v_i,y-1/2)} \otimes m_{(v_i,y+1/2)}. \tag{14}$$

To summarize, while the fusion rules of electric and magnetic charges in the $y$-direction have the same form as the toric code in that a pair of fractional excitations are created, in the $x$-direction, the fusion rules show unusual behavior and their form crucially depends on the Laplacian. In the next subsection, using this property, we will count the number of superselection sectors in our model.

## 3.2 Superselection sectors and GSD

In this subsection, we study how many distinct fractional excitations in our model, more precisely, we count the number of superselection sectors. In doing so, we impose periodic boundary condition in the $y$-direction by identifying the coordinate $(v_i, y)$ with $(v_i, y+n_y)$ $(n_y \in \mathbb{Z})$.

The number of superselection sectors in our model amounts to be the number of distinct closed loop of fractional excitations that goes around the system. For instance, in the case

of the $\mathbb{Z}_2$ toric code, the superselection sectors are labeled by distinct non-contractible closed loops of electric and magnetic charges going around in either $x$- or $y$-direction [8].

We apply the same logic to the present case. We first focus on the closed loops of electric charges in the $x$-direction. Assuming the graph $G(V, E)$ at given $y$ has $n$ vertices, we consider a closed loop of an electric charge in the form $Z^{r_1}_{1,(v_1,y)} Z^{r_2}_{1,(v_2,y)} \cdots Z^{r_n}_{1,(v_n,y)}$ characterized by the vector $\mathbf{r} \in \mathbb{Z}_N^n$ defined in (8). The loop must satisfy that they commute with every term of $V_{(v_i,y)}$ given in (4), i.e. the loop does not violate $V_{(v_i,y)} = 1$ at any coordinate. In other words, the fusion rule of electric charges induced by applying the operator $Z^{r_1}_{1,(v_1,y)} Z^{r_2}_{1,(v_2,y)} \cdots Z^{r_n}_{1,(v_n,y)}$ on the ground state becomes trivial. Recalling the argument around (8)-(11), and using the Laplacian $L$, this condition is equivalent to

$$L\mathbf{r} = \mathbf{0} \mod N. \tag{15}$$

Therefore, the number of closed loops of electric charge in the $x$-direction is associated with the kernel of the Laplacian, $\ker(L)$. Note that there are always at least $N$ solutions of (15) $\mathbf{r} = k(1, 1, \cdots, 1)^T$ ($k \in \mathbb{Z}_N$), mirroring the fact that the Laplacian is singular, meaning summing over matrix elements along row gives zero. As we will see soon, depending on the Laplacian, there can be more than $N$ solutions.

To proceed, we transform the Laplacian into the Smith normal form (2). By multiplying invertible matrices $P$ and $Q$ over integers on the Laplacian, the condition (15) becomes

$$P^{-1}DQ^{-1}\mathbf{r} = 0 \mod N \iff D\tilde{\mathbf{r}} = 0 \mod N, \tag{16}$$

where in the second equality, we have multiplied $P$ from the left and used the fact that $P$ is a matrix over integers. Also, we have introduced $\tilde{\mathbf{r}} := Q^{-1}\mathbf{r} \in \mathbb{Z}_N^n$ (Note that $P^{-1}$ and $Q^{-1}$ are also integer matrices.). Suppose the Smith normal form of the Laplacian (2) has $m$ invariant factors greater than one, meaning

$$D = \mathrm{diag}(\underbrace{1, \cdots, 1}_{n\text{-}1\text{-}m}, \underbrace{p_1, \cdots, p_m}_{m}, 0), \quad (p_i \geq 2). \tag{17}$$

Then, from (16), it follows that the first $n - 1 - m$ components of the vector $\tilde{\mathbf{r}}$ are zero:

$$\tilde{r}_i = 0 \mod N \, (1 \leq i \leq n - 1 - m). \tag{18}$$

As for the elements $\tilde{r}_{i+n-1-m}$ ($1 \leq i \leq m$), they have to satisfy

$$p_i \tilde{r}_{i+n-1-m} = 0 \mod N \iff p_i \tilde{r}_{i+n-1-m} = N t_i \, (1 \leq i \leq m, \, t_i \in \mathbb{Z}). \tag{19}$$

Decompose $N$ and $p_i$ into two integers as

$$N = N_i' \gcd(N, p_i), \quad p_i = p_i' \gcd(N, p_i), \tag{20}$$

where gcd stands for the greatest common divisor and $N_i'$ and $p_i'$ are coprime, (19) becomes $p_i' \tilde{r}_{i+n-1-m} = N_i' t_i$. Since $N_i'$ and $p_i'$ are coprime, one finds

$$\tilde{r}_{i+n-1-m} = N_i' \alpha_i, \quad (1 \leq i \leq m), \tag{21}$$

where integer $\alpha_i$ can take $\gcd(N, p_i)$ distinct values, i.e. $\alpha_i = 0, 1, \cdots, \gcd(N, p_i) - 1$. There is no constraint on the last element of $\tilde{\mathbf{r}}$, $\tilde{r}_N$ as the last diagonal entry of $D$ is zero, implying $\tilde{r}_N$ can take $N$ distinct values.

Overall, with the assumption of (17), the condition (16) leads to that

$$\tilde{\mathbf{r}} = (\underbrace{\tilde{r}_1, \cdots, \tilde{r}_{n-1-m}}_{n\text{-}1\text{-}m}, \underbrace{\tilde{r}_{n-m}, \cdots, \tilde{r}_{n-1}}_{m}, \tilde{r}_n)^T = (\underbrace{0, \cdots, 0}_{n\text{-}1\text{-}m}, \underbrace{N_1' \alpha_1, \cdots, N_m' \alpha_m}_{m}, \alpha')^T \mod N, \tag{22}$$

where $0 \leq \alpha_i \leq \gcd(N, p_i) - 1$, $0 \leq \alpha' \leq N - 1$. Hence, the kernel of the Laplacian, associated with closed loops of electric charges, is labeled by

$$\mathbb{Z}_N \times \mathbb{Z}_{\gcd(N,p_1)} \times \mathbb{Z}_{\gcd(N,p_2)} \times \cdots \times \mathbb{Z}_{\gcd(N,p_m)}. \tag{23}$$

When $\gcd(N, p_i) = 1$, the sector $\mathbb{Z}_{\gcd(N,p_i)}$ becomes trivial as $\alpha_i$ takes trivial value, i.e. $\alpha_i = 0$ mod $N$.

Once we have identified the closed loops of electric charges in the $x$-direction, they can be deformed into the ones shifted upwards or downwards in the $y$-direction by applying sets of $P_{(v_i,y+1/2)}$, corresponding to the fusion rule (12), therefore, we have exhausted all kinds of closed loops of electric charges in the $x$-direction, fully labeled by (23). The explicit form of the closed loops, consisting of multiple of $Z_1$ operators, is obtained by multiplying the matrix $Q$ from the left in (22). More explicitly, the closed loop of an electric charge in the $x$-direction, $W_{e,x,\mathbf{r}_\alpha}$ is labeled by $\alpha := (\mathbb{Z}_{\gcd(N,p_1)}, \cdots, \mathbb{Z}_{\gcd(N,p_m)}, \mathbb{Z}_N)$ via

$$W_{e,x,\mathbf{r}_\alpha} = Z_{1,(v_1,y)}^{r_1} \cdots Z_{n,(v_n,y)}^{r_n}, \quad \mathbf{r} = QV \begin{pmatrix} \mathbf{0}_{n-m-1} \\ \alpha \end{pmatrix} \mod N. \tag{24}$$

Here,

$$V = \operatorname{diag}(\underbrace{1, \cdots, 1}_{n\text{-}1\text{-}m}, \underbrace{N_1', \cdots, N_m'}_{m}, 1), \tag{25}$$

where $N_i'$ is defined in (20), and $\mathbf{0}_{n-m-1}$ denotes $n-m-1$ dimensional vector with all entries being zero.

The closed loops of magnetic charges in the $x$-direction can be discussed in the similar fashion, leading to that they carry the same quantum numbers (23), thus, we finally find that the superselection sectors are characterized by $\left[\mathbb{Z}_N \times \mathbb{Z}_{\gcd(N,p_1)} \times \mathbb{Z}_{\gcd(N,p_2)} \times \cdots \times \mathbb{Z}_{\gcd(N,p_m)}\right]^2$. The multiple ground states are distinguished by the closed loops of electric and magnetic charges in the $x$-direction, therefore, we have

$$\boxed{\mathrm{GSD} = \left[N \times \gcd(N, p_1) \times \cdots \times \gcd(N, p_m)\right]^2.} \tag{26}$$

The GSD dependence on gcd of $N$ and length of the lattice was studied in the Wen's $\mathbb{Z}_2$ plaquette model and several models in the case of the square lattice [21,22]. Here, by employing algebraic tools of graph theory, we have derived such novel GSD dependence in the case of an arbitrary connected graph.

### 3.3 Alternative derivation of (26)

There is an alternative approach to reach (26) by counting the number of superselection sectors of electric and magnetic charges in the $y$-direction, instead of the $x$-direction. Let us first focus on the case with magnetic charges. As we mentioned in the previous subsection, the fusion rules of the magnetic charges in the $y$-direction (14) is identical to the one in the toric code, where a pair of excitations are created. From this feature, one can construct a closed loop of magnetic charge formed by string of $X_1$ operators defined by

$$W_{m,v_i} := \prod_{y=1}^{n_y-1} X_{1,(v_i,y)}. \tag{27}$$

Although this loop resembles the one in the toric code, there is a crucial difference between the two. Depending on $N$ and the graph, one fails to deform the loop by applying sets of operators $V_{(v_i,y)}$ so that it is shifted to the adjacent position in the $x$-direction. Rather, the

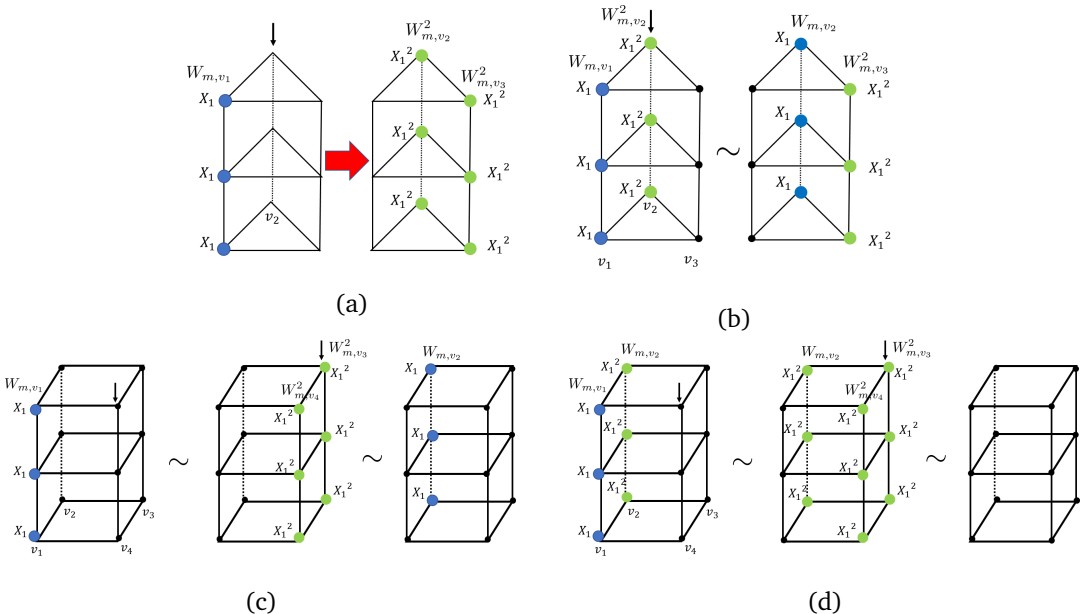

Figure 3: Deformation of the closed loops of magnetic charges in the $y$-direction in the case where the graph is the cycle graph $C_3$ or $C_4$ with $N = 3$. The top (bottom) figures correspond to the cycle graph $C_3$ ($C_4$). The deformation is implemented by acting $\prod_y V^2_{(v_i,y)}$ (along the vertical line marked by the black arrow) on the loop. The symbol "$\sim$" represents identification between two configurations under the deformation.

loop is deformed into composite of loops, $W_{m,v_i}$ and $W_{m,v_j}$ with $v_j$ being adjacent vertex in the $x$-direction.

To understand what we have just mentioned more intuitively, let us for a moment consider the case where the graph is the cycle graph $C_n$ and $N = 3$ as shown in Fig 3. For simplicity, we concentrate on the two cases with $C_3$ and $C_4$. In the case of $C_3$, by acting $\prod_y V^2_{(v_2,y)}$ on a single closed loop $W_{m,v_1}$ (For the sake of simplicity, $\prod_{y=1}^{n_y-1}$ is abbreviated as $\prod_y$.), it is deformed into the composite of $W^2_{m,v_2}$ and $W^2_{m,v_3}$ (Fig. 3a). After some trials, one is convinced that the single loop cannot be shifted to the adjacent position under any deformation. However, if we start with a "dipole" of the closed loops, $W_{m,v_1} W^2_{m,v_2}$, it can be shifted to the adjacent position in the $x$-direction under the deformation as shown in Fig. 3b, thus it moves around the system. This property reminds us of topological defects in smectic phase in a liquid crystal, where a dipole of excitations which is dislocation, is free to move in one direction whereas a single one, disclination cannot [23]. On the contrary, in the case of $C_4$, the single loop can be shifted to the adjacent position under the deformation, as portrayed in Fig. 3c, where a single loop $W_{m,v_1}$ is shifted to $W_{m,v_2}$. Also, there is no dipole configuration – it becomes vacuum configuration under the deformation (Fig. 3d). As we will see below, whether the single loop can be shifted or the phase admits the dipole of closed loops depends on $N$ and invariant factors of the Laplacian.

Coming back to the generic case of the graph, in order to count the superselection sectors coming from the magnetic charges, one has to know the number of distinct configurations of the loops up to deformation. From (4), after simple algebra, one finds

$$\prod_y V_{(v_i,y)} = W^{-\deg(v_i)}_{m,v_i} \prod_j W^{l_{ij}}_{m,v_j}.$$  (28)

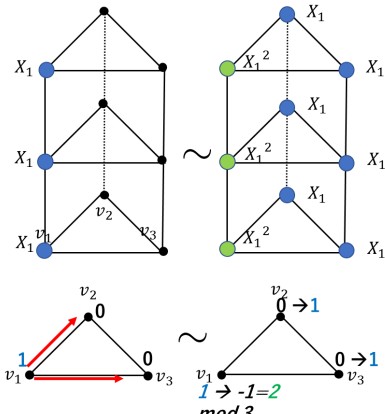

Figure 4: Deformation of closed loops of magnetic charges in the case of the cycle graph $C_3$ and $N = 3$. (Top) Deformation of $W_{m,v_1}$ by $\prod_y V_{(v_1,y)}$. (Bottom) The top view of our lattice, where one assigns $\mathbb{Z}_3$ number on each vertex, corresponding to the configuration of the loops. These numbers are regarded as chips located at each vertex. By applying $\prod_y V_{(v_1,y)}$, the closed loop is deformed, which corresponds to the one of the chip-firing process where the chip at vertex $v_1$ is transferred into the adjacent ones, $v_2$ and $v_3$ (red arrows).

By acting this operator on a closed loop $W_{m,v_i}$, it yields

$$\prod_y V_{(v_i,y)} W_{m,v_i} = W_{m,v_i}^{1-\deg(v_i)} \prod_j W_{m,v_j}^{l_{ij}}. \tag{29}$$

This consideration becomes more succinct in the language of the Laplacian. We define a vector $\mathbf{s} \in \mathbb{Z}_N^n$ associated with sets of closed loops, $W_{m,v_1}^{s_1} \times \cdots \times W_{m,v_n}^{s_n}$. Also, introducing another vector $\sigma \in \mathbb{Z}_N^n$, corresponding to $\prod_y V_{(v_1,y)}^{\sigma_1} \times \cdots \times \prod_y V_{(v_n,y)}^{\sigma_n}$, which acts on the sets of the closed loops. The configuration of the closed loops after the deformation becomes

$$\mathbf{s} - L\sigma (:= \mathbf{s}'). \tag{30}$$

The number of distinct configurations of the closed loops is equivalent to the number of distinct $\mathbf{s}$ under the identification $\mathbf{s} \sim \mathbf{s}'$. Therefore, we need to find $\mathbb{Z}_N^n/\mathrm{im}(L)$, which is known as the *Picard group*, $\mathrm{Pic}(G)$.

What we have discussed so far in this subsection has intimate relation to the *chip-firing game*, invented in the context of graph theory [24–26]. This interpretation becomes clearer from the top view of our lattice (Fig. 4). The configuration of the closed loops of magnetic charges, $\mathbf{s} \in \mathbb{Z}_N^n$ corresponds to what is called chip configuration in the chip-firing game which is defined as non-negative integer vector recording the number of chips located at each vertex of the graph and the process of the deformation of the closed loops can be regarded as the process of chip-firing where one chip is sent to each of its neighbors. In the chip-firing game, the Picard group is studied to classify the configurations of chips [24–26]. Important difference between the chip-firing game and our consideration is that while a chip takes non-negative integer at each vertex in the chip-firing game, $\mathbb{Z}_N$ number is assigned at each vertex in our case.

To find the Picard group, we need to evaluate $\mathrm{im}(L)$. To this end, remembering the Lapla-

cian is transformed into the Smith normal form (2), we have

$$
\begin{aligned}
\mathrm{im}(L) &= L\sigma, \ \forall \sigma \in \mathbb{Z}_N^n \\
&= P^{-1}D\tilde{\sigma} \ (\tilde{\sigma} := Q^{-1}\sigma) \\
&= \mathrm{span}(\pi_1', \pi_2', \cdots, \pi_n').
\end{aligned} \tag{31}
$$

Here, $\pi_i'$ represents the vector corresponding to the $i$-th column of $P^{-1}D$. Since $D$ is the diagonal with the last entry being zero, (31) is further written as

$$
\mathrm{im}(L) = \mathrm{span}(u_1\pi_1, u_2\pi_2, \cdots, u_{n-1}\pi_{n-1}), \tag{32}
$$

where $\pi_i$ denotes the vector which corresponds to the $i$-th column of $P^{-1}$. Now we write $\mathbf{s} \in \mathbb{Z}_N^n / \mathrm{im}(L)$ in these basis:

$$
\mathbf{s} = \sum_{i=1}^n c_i \pi_i, \quad (c_i \in \mathbb{Z}_N). \tag{33}
$$

From (32), $c_i$ is subject to (the symbol "$\sim$" represents identification)

$$
c_i \sim c_i + u_i \ (1 \le i \le n-1). \tag{34}
$$

By definition, it also must satisfy

$$
c_i \sim c_i + N \ (1 \le i \le n). \tag{35}
$$

The algebraic structure of the Picard group is determined by the number of distinct $\mathbf{s}$ subject to the two constraints (34)(35). Assuming the Smith normal form of the Laplacian has $m$ invariant factors greater than one as shown in (17), then we have

$$
c_i \sim c_i + 1 \ (1 \le i \le n-1-m),
$$

implying the coefficients of the first $n-1-m$ basis are trivial. As for the coefficients $c_{i+n-1-m}$ $(1 \le i \le m)$, they satisfy

$$
\begin{aligned}
c_{i+n-1-m} &\sim c_{i+n-1-m} + p_i, \\
c_{i+n-1-m} &\sim c_{i+n-1-m} + N \ (1 \le i \le m),
\end{aligned}
$$

which leads to that $c_{i+n-1-m}$ $(1 \le i \le m)$ can take $\gcd(N, p_i)$ distinct values. Together with the fact that the last coefficient $c_n$ can take $N$ distinct values, we find that

$$
\mathbf{c} := (\underbrace{c_1, \cdots, c_{n-1-m}}_{n\text{-}1\text{-}m}, \underbrace{c_{n-m}, \cdots, c_{n-1}}_{m}, c_n)^T = (\underbrace{0, \cdots, 0}_{n\text{-}1\text{-}m}, \underbrace{\beta_1, \cdots, \beta_m}_{m}, \beta')^T \mod N, \tag{36}
$$

with $\beta_i \in \mathbb{Z}_{\gcd(N,p_i)}$, $\beta' \in \mathbb{Z}_N$. Therefore, the distinct configurations of the closed loops of magnetic charges are labeled by

$$
\mathbb{Z}_N \times \mathbb{Z}_{\gcd(N,p_1)} \times \mathbb{Z}_{\gcd(N,p_2)} \times \cdots \times \mathbb{Z}_{\gcd(N,p_m)}.
$$

Since

$$
\mathbf{s} = \sum_{i=1}^n c_i \pi_i = P^{-1}\mathbf{c}, \tag{37}
$$

the explicit form of the configuration of the loops $\mathbf{s}$ is obtained by multiplying $P^{-1}$ from the left in (36). To be more precise, the closed loops of magnetic charges running in the $y$-direction, $W_{m,y,\mathbf{s}_\beta}$ is labeled by $\beta := (\mathbb{Z}_{\gcd(N,p_1)}, \cdots, \mathbb{Z}_{\gcd(N,p_m)}, \mathbb{Z}_N)$ via

$$
W_{m,y,\mathbf{s}_\beta} = W_{m,v_1}^{s_1} \times \cdots \times W_{m,v_n}^{s_n}, \quad \mathbf{s} = P^{-1}\begin{pmatrix} \mathbf{0}_{n-m-1} \\ \beta \end{pmatrix} \mod N. \tag{38}
$$

One can similarly discuss the configurations of the closed loops of electric charges in the $y$-direction, giving the same result as the case with the magnetic charges. Hence, the GSD is given by (26).

### 3.4 Braiding statistics

Based on discussions presented in preceding subsections, one can evaluate braiding statistics between electric and magnetic charges. In dosing so, we make use of the analogous logic to obtain the statistics in the toric code. In the case of the $\mathbb{Z}_2$ toric code, the non-trivial braiding statistics is characterized by $\theta$ via $W_{e,x}W_{m,y} = e^{i\theta}W_{m,y}W_{e,x}$, where $W_{e,x}(W_{e,y})$ represents the closed loop of electric (magnetic) charge in the $x(y)$-direction and the phase factor is given by $\theta = \pi$.

In our model, we have closed loops of fractional excitations in the $x$-direction and the ones in the $y$-direction, both of which are labeled by $\alpha, \beta \in \prod_{i=1}^{m} \mathbb{Z}_{\gcd(N,p_i)} \times \mathbb{Z}_N$. From (3), (24), (25), and (38), introducing sub-diagonal $(m+1) \times (m+1)$ matrix of $(P^{-1})^T VQ$ by $\Gamma$ via

$$(P^{-1})^T VQ = \begin{pmatrix} * & * \\ * & \Gamma \end{pmatrix}, \tag{39}$$

where the symbol "$*$" denotes some matrix element which is not necessary to find in the present discussion, the statistical relation between the two loops is described by

$$\boxed{W_{e,x,\mathbf{r}_\alpha} W_{m,y,\mathbf{s}_\beta} = \omega^{\beta^T \Gamma \alpha} W_{m,y,\mathbf{s}_\beta} W_{e,x,\mathbf{r}_\alpha}.} \tag{40}$$

The braiding statistics between a closed loop of a magnetic charge in the $x$-direction and the one of an electric charge in the $y$-direction has the same form as (40).

Generally, the form of the matrices $P$ and $Q$ is not uniquely determined, mirroring the fact that there are multiple ways to transforming the Laplacian into the Smith normal form. In our context, such a fact corresponds to the base transformation of the superselection sectors, retaining physical properties of the system, such as spectrum. More thorough analysis on statistics of fractional excitations will be presented elsewhere.

### 3.5 More generic 2D lattice

We can extend our analysis to the case where the 2D phases are placed on more generic 2D lattices constructed by product of two connected graphs, $G_1(V_1, E_1) \otimes G_2(V_2, E_2)$. Practically, such lattices are obtained by replacing the 1D line (the line in $y$-direction) introduced in Sec. 2 with a connected graph $G_2(V_2, E_2)$ and relabeling the graph in the $x$-direction as $G_1(V_1, E_1)$. Generalization of Hamiltonian (5) to such lattices is straightforward. One of examples of such lattices is portrayed in Fig. 5, consisting of the cyclic group $C_2$ and a tree.

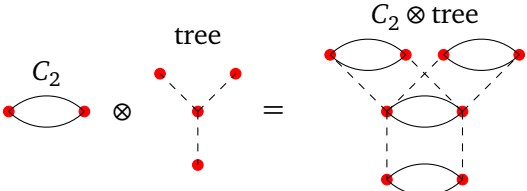

Figure 5: An example of a 2D lattice as a product of two connected graphs. The red dots are nodes of the lattice. Solid and dashed lines are the edges along the horizontal and vertical directions, respectively.

As we previously mentioned, in the $y$-direction, the behavior of the excitations closely parallels the one in the toric code. Thus, properties of the excitations in the $y$-direction only depends on the global topology of the graph, not the Laplacian. In deriving the GSD by the second approach (Sec. 3.3), we initially constructed a single closed loop of a magnetic charge

in the $y$-direction. If we instead consider the case of the 2D lattice where 1D line is replaced with graph $G_2(V_2, E_2)$, there are $g_2$ ways to form such a closed loop with $g_2$ being the genus of graph $G_2$, $g_2 := |E_2| - |V_2| + 1$. In other words, additional $g_2$ degrees of freedom is assigned to each closed loop of a magnetic charge. Accordingly, one finds that the distinct configurations of closed loops of magnetic charges are labeled by

$$\left[ \mathbb{Z}_N \times \mathbb{Z}_{\gcd(N,p_1)} \times \mathbb{Z}_{\gcd(N,p_2)} \times \cdots \times \mathbb{Z}_{\gcd(N,p_m)} \right]^{g_2}. \tag{41}$$

Taking closed loops of electric charges into the consideration [these are also labeled by (41)], one arrives at

$$\boxed{\mathrm{GSD} = \left[ N \times \gcd(N, p_1) \times \cdots \times \gcd(N, p_m) \right]^{2g_2}, \quad g_2 = |E_2| - |V_2| + 1.} \tag{42}$$

## 4 Examples

In this, section, we examine two simple examples of graph to see explicitly how our results (26) (40) work.

### 4.1 Cycle graph

The cycle graph $C_n$ consists of $n$ vertices placed in a cyclic order so that adjacent vertices are connected by a single edge. The 2D lattice construed from the product of the cycle graph and 1D line with periodic boundary condition in the $y$-direction is equivalent to the torus geometry with periodicity in the $x$-direction being $n$. We transform the Laplacian to the Smith normal form by implementing operations on rows and columns.

To start, adding the first $n-1$ columns to the last one and doing the same procedure for rows, the Laplacian is transformed as

$$L = \begin{pmatrix} 2 & -1 & & & & -1 \\ -1 & 2 & -1 & & & \\ & -1 & 2 & \ddots & & \\ & & & \ddots & \ddots & -1 \\ -1 & & & & -1 & 2 \end{pmatrix} \rightarrow \begin{pmatrix} \tilde{L} & \mathbf{0}_{n-1} \\ \mathbf{0}_{n-1}^T & 0 \end{pmatrix}, \tag{43}$$

where

$$\tilde{L} = \begin{pmatrix} 2 & -1 & & & \\ -1 & 2 & -1 & & \\ & -1 & 2 & \ddots & \\ & & \ddots & \ddots & -1 \\ & & & -1 & 2 \end{pmatrix}_{n-1 \times n-1}. \tag{44}$$

The Laplacian of any connected graph is transformed into the form (43), where $\tilde{L}$ is obtained by removing the last row and column of the Laplacian. We further transform $\tilde{L}$ as

$$\tilde{L} \xrightarrow[\text{and negate on } 1^{st} \text{ row}]{\text{swap } 1^{st} \text{ and } 2^{nd} \text{ row}} \begin{pmatrix} 1 & -2 & 1 & & \\ 2 & -1 & & & \\ & -1 & 2 & \ddots & \\ & & \ddots & \ddots & -1 \\ & & & -1 & 2 \end{pmatrix} \xrightarrow[\text{and subtract } 1^{st} \text{ column from } 3^{rd} \text{ one}]{\text{add } 1^{st} \text{ column to } 2^{nd} \text{ one twice}} \begin{pmatrix} 1 & 0 & 0 & & \\ 2 & 3 & -2 & & \\ & -1 & 2 & \ddots & \\ & & \ddots & \ddots & -1 \\ & & & -1 & 2 \end{pmatrix}. \tag{45}$$

Continuing,

$$(45) \xrightarrow{\text{subtract } 1^{st} \text{ row from } 2^{nd} \text{ one twice}} \begin{pmatrix} 1 & 0 & 0 & & & \\ 0 & 3 & -2 & & & \\ & -1 & 2 & \ddots & & \\ & & & \ddots & \ddots & -1 \\ & & & & -1 & 2 \end{pmatrix}. \tag{46}$$

The last form of (46) has a diagonal element in $(1,1)$ entry. We iteratively implement the similar transformation on the sub-diagonal matrix below $(1,1)$ entry by swapping the first and second rows of the sub-diagonal matrix followed by multiplying $(-1)$ on the first row, and adding the first columns and rows to or subtracting those from other columns and rows. Finally, one arrives at

$$PLQ = \text{diag}(1, 1, \cdots, n, 0), \tag{47}$$

where matrix $P$ ($Q$) corresponds to the operations involving switching between rows (columns), negating, and adding or subtracting the rows (columns). From the Smith normal form (47), there is only one invariant factor greater than one, which is the second diagonal element from the last, $n$. Applying formula (26) to the present case, one finds

$$\text{GSD} = [N \times \gcd(N, n)]^2. \tag{48}$$

From our operations on columns and rows performed in (43)-(46), one can find $P^{-1}$ and $Q$ as

$$P^{-1} = \begin{pmatrix} 2 & 3 & \cdots & n-1 & 1 & 0 \\ -1 & & & & & 0 \\ & -1 & & & & \vdots \\ & & \ddots & & & \vdots \\ & & & -1 & & 0 \\ -1 & -2 & \cdots & -(n-2) & -1 & 1 \end{pmatrix}, \quad Q = \begin{pmatrix} 1 & 2 & 3 & \cdots & n-1 & 1 \\ & 1 & 2 & \cdots & n-2 & 1 \\ & & \ddots & \ddots & & \vdots \\ & & & 1 & 2 & 1 \\ & & & & 1 & 1 \\ & & & & & 1 \end{pmatrix}. \tag{49}$$

The superselection sectors are labeled by $\mathbb{Z}_{\gcd(N,n)} \times \mathbb{Z}_N$. From (49), one finds that the form of the closed loop of electric charge in the $x$-direction, $W_{e,x,\mathbf{r}_\alpha}$, labeled by $\alpha = (\alpha_1, \alpha')^T \in \mathbb{Z}_{\gcd(N,n)} \times \mathbb{Z}_N$, is described by Eqs. (24)(25) with

$$\mathbf{r} = QV \begin{pmatrix} \mathbf{0}_{n-2} \\ \alpha_1 \\ \alpha' \end{pmatrix} = N_1' \alpha_1 \begin{pmatrix} n-1 \\ n-2 \\ \vdots \\ 1 \\ 0 \end{pmatrix} + \alpha' \begin{pmatrix} 1 \\ 1 \\ \vdots \\ 1 \\ 1 \end{pmatrix} \quad \text{mod } N, \tag{50}$$

where $N_1' = N/\gcd(N, n)$. The last term of (50) corresponds to the solution of $L\mathbf{r} = \mathbf{0}$ that any connected graph has. Similarly, refereeing to (49), the configuration of the closed loop of magnetic charge in the $y$-direction, $W_{m,y,\mathbf{s}_\beta}$ characterized by $\beta = (\beta_1, \beta')^T \in \mathbb{Z}_{\gcd(N,n)} \times \mathbb{Z}_N$ has the form (38) with

$$\mathbf{s} = P^{-1} \begin{pmatrix} \mathbf{0}_{n-2} \\ \beta_1 \\ \beta' \end{pmatrix} = \beta_1 \begin{pmatrix} 1 \\ 0 \\ \vdots \\ 0 \\ -1 \end{pmatrix} + \beta' \begin{pmatrix} 0 \\ 0 \\ \vdots \\ 0 \\ 1 \end{pmatrix} \quad \text{mod } N. \tag{51}$$

When $\gcd(n, N) \neq 1$, the phase admits dipole of closed loops of magnetic charges in accordance with the first term of (51). In Fig. 6, we demonstrate configurations of closed loops of electric

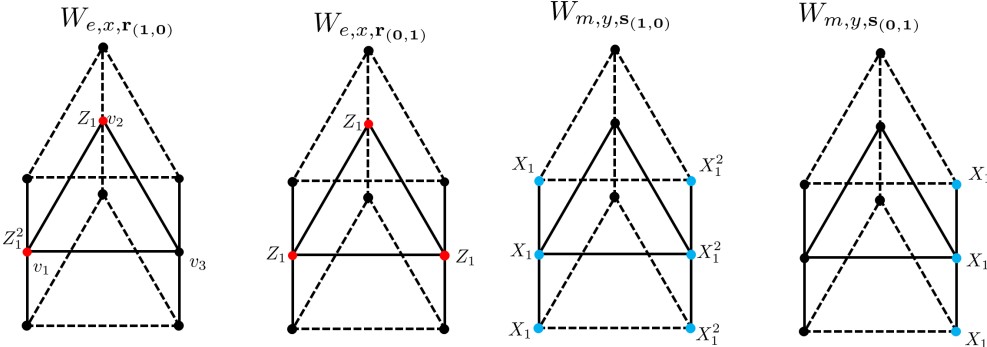

Figure 6: Two configurations of the closed loops of electric charge in the $x$-direction (24) with $\alpha = (1,0),(0,1)$ (red dots) and those of magnetic charges in the $y$-direction (38) with $\beta = (1,0),(0,1)$ (blue dots) in the case of the cycle graph $C_n$ with $N = n = 3$.

charges with $\alpha = (1,0),(0,1)$ and those of magnetic charges with $\beta = (1,0),(0,1)$ in the case of $N = n = 3$, from which any configuration of loops is constructed. By calculating $(P^{-1})^T V Q$, the matrix $\Gamma$ given in (39), the braiding statistics between $W_{e,x,\mathbf{r}_\alpha}$ and $W_{m,y,\mathbf{s}_\beta}$ reads

$$W_{e,x,\mathbf{r}_\alpha} W_{m,y,\mathbf{s}_\beta} = \omega^{\beta^T \Gamma \alpha} W_{m,y,\mathbf{s}_\beta} W_{e,x,\mathbf{r}_\alpha}, \quad \text{with } \Gamma = \begin{pmatrix} N_1'(n-1) & \\ & 1 \end{pmatrix}. \tag{52}$$

## 4.2 Complete graph

We move on to another example, complete graph $K_n$. It comprised of $n$ vertices, where there is an edge between any pair of vertices. The Laplacian is described by

$$L = \text{diag}(n,n,n,\cdots,n) - A, \tag{53}$$

where $A$ represents the all-ones matrix (i.e. the matrix with all entries being one). To transform the Laplacian into the Smith normal form, we add the first $n-1$ columns to the last one and implementing the same manipulation for the rows, giving the form

$$L \to \begin{pmatrix} \tilde{L} & \mathbf{0}_{n-1} \\ \mathbf{0}_{n-1}^T & 0 \end{pmatrix},$$

with $\tilde{L}$ being the matrix obtained by removing the last row and column of $L$. We further transform $\tilde{L}$. Adding all of the second to the last rows to the first one, and adding the first column to all of the second to the last columns, we have

$$\tilde{L} \to \begin{pmatrix} 1 & 0 & \cdots & & 0 \\ 1 & n & & & \vdots \\ \vdots & & \ddots & & 0 \\ 1 & & & & n \end{pmatrix}.$$

Subtracting the first rows from all other rows gives rise to the Smith normal form:

$$PLQ = \text{diag}(1,n,n,\cdots,n,0). \tag{54}$$

There are $n-2$ invariant factors greater than one, all of which are $n$. By making use of (26), we obtain

$$\text{GSD} = \left[ N \times (\gcd(N,n))^{n-2} \right]^2. \tag{55}$$

One of possible form of the matrices $P^{-1}$ and $Q$, corresponding to transformations on columns and rows mentioned above, is given by

$$P^{-1} = \begin{pmatrix} 1 & & & & \\ -1 & 1 & & & \\ \vdots & & \ddots & & \\ -1 & & & 1 & \\ n-3 & -1 & -1 & -1 & 1 \end{pmatrix}, \quad Q = \begin{pmatrix} 1 & 1 & \cdots & \cdots & 1 \\ 1 & 2 & 1 & \cdots & 1 \\ \vdots & 1 & \ddots & & \vdots \\ 1 & 1 & 1 & 2 & 1 \\ 0 & 0 & 0 & 0 & 1 \end{pmatrix}. \tag{56}$$

The superselection sectors are characterized by $\mathbb{Z}^{n-2}_{\gcd(N,n)} \times \mathbb{Z}_N$. The closed loop of electric charge in the $x$-direction, given in (24) and (25), is labeled by $\alpha = (\alpha_1, \alpha_2, \cdots, \alpha_{n-2}, \alpha') \in \mathbb{Z}^{n-2}_{\gcd(N,n)} \times \mathbb{Z}_N$ with

$$\mathbf{r} = QV \begin{pmatrix} \mathbf{0} \\ \alpha \end{pmatrix} = N'\alpha_1 \begin{pmatrix} 1 \\ 2 \\ 1 \\ \vdots \\ \vdots \\ 1 \\ 0 \end{pmatrix} + N'\alpha_2 \begin{pmatrix} 1 \\ 1 \\ 2 \\ 1 \\ \vdots \\ 1 \\ 0 \end{pmatrix} + \cdots + N'\alpha_{n-2} \begin{pmatrix} 1 \\ 1 \\ 1 \\ \vdots \\ 1 \\ 2 \\ 0 \end{pmatrix} + \alpha' \begin{pmatrix} 1 \\ 1 \\ 1 \\ \vdots \\ 1 \\ 1 \\ 1 \end{pmatrix} \quad \text{mod } N, \tag{57}$$

where $N' = N/\gcd(N,n)$. Likewise, closed loops of magnetic charges in the $y$-direction, defined in (38), which are labeled by $\beta = (\beta_1, \beta_2, \cdots, \beta_{n-2}, \beta') \in \mathbb{Z}^{n-2}_{\gcd(N,n)} \times \mathbb{Z}_N$ with

$$\mathbf{s} = P^{-1} \begin{pmatrix} \mathbf{0} \\ \beta \end{pmatrix} = \beta_1 \begin{pmatrix} 0 \\ 1 \\ 0 \\ \vdots \\ \vdots \\ 0 \\ -1 \end{pmatrix} + \beta_2 \begin{pmatrix} 0 \\ 0 \\ 1 \\ 0 \\ \vdots \\ 0 \\ -1 \end{pmatrix} + \cdots + \beta_{n-2} \begin{pmatrix} 0 \\ 0 \\ 0 \\ \vdots \\ 0 \\ 1 \\ -1 \end{pmatrix} + \beta' \begin{pmatrix} 0 \\ 0 \\ 0 \\ \vdots \\ 0 \\ 0 \\ 1 \end{pmatrix} \quad \text{mod } N. \tag{58}$$

We portray configurations of (57) and (58) in Fig. 7 with $n = N = 4$ where $\alpha$ and $\beta$ take the form of the standard basis of vector e.g., $\alpha = (1, 0, \cdots, 0)^T$.

The braiding statistics between the closed loop of electric and the one of magnetic charge is given by

$$W_{e,x,\mathbf{r}_\alpha} W_{m,y,\mathbf{s}_\beta} = \omega^{\beta^T \Gamma \alpha} W_{m,y,\mathbf{s}_\beta} W_{e,x,\mathbf{r}_\alpha}, \quad \text{with } \Gamma = \begin{pmatrix} 2N' & N' & \cdots & \cdots & N' & 0 \\ N' & 2N' & N' & \cdots & N' & 0 \\ \vdots & N' & \ddots & & \vdots & \vdots \\ \vdots & \vdots & & & N' & \\ N' & N' & \cdots & N' & 2N' & 0 \\ 0 & \cdots & & & 0 & 1 \end{pmatrix}. \tag{59}$$

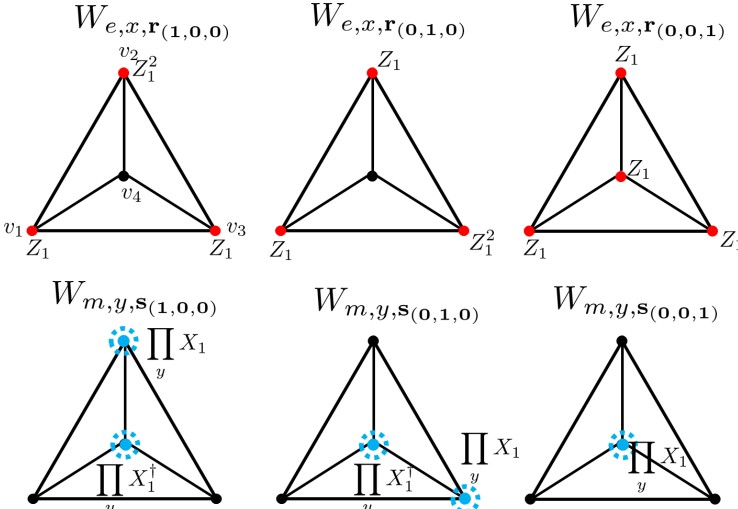

Figure 7: The top view of three configurations of the closed loops of electric charge in the $x$-direction (24) with $\alpha = (1, 0, 0), (0, 1, 0), (0, 0, 1)$ (red dots) and those of magnetic charges in the $y$-direction (38) with $\beta = (1, 0, 0), (0, 1, 0), (0, 0, 1)$ (blue dots) in the case of the complete graph $K_n$ with $n = N = 4$. The blue dot with dashed circle represents the closed loop of the magnetic charge which is directed out of the paper.

## 5 Conclusion

Spurred by a recent discovery of the fracton topological phases, in this paper, we have studied unusual gapped $\mathbb{Z}_N$ topological phases on 2D lattice which is constructed by the product of an arbitrary connected graph and 1D line, and explored interplay between fractional excitations and combinatorics. The distinct property of our model from the conventional topologically ordered phases is that depending on $N$ and the graph, the world line of a fractional excitation cannot be deformed to shift to the adjacent position in the $x$-direction. Rather, the composite of world lines of the excitations can be shifted. Such a property can be seen more clearly in the case of the square lattice by setting $G(V, E)$ to be the cycle graph $C_n$ with $\gcd(N, n) \neq 1$, where dipole excitations are free to move. This behavior is reminiscent of topological defects of the smectic phase in a liquid crystal, where dipoles of disclinations move freely in one direction [23].

Due to this mobility constraint, the model exhibits unusual GSD dependence on $N$ and the graph. We have derived the GSD dependence on the graph (26) by two approaches. In the first approach, we have shown that the superselection sectors are characterized by the kernel of the Laplacian. By the knowledge of graph theory, we have found that GSD depends on $N$ and the great common divisor of $N$ and invariant factors of the Laplacian. In the second approach, we evaluate the number of distinct configurations of closed loops of fractional excitations in the $y$-direction up to the deformation. Finding an intriguing analogy between our model and the chip-firing game, we have obtained the same GSD dependence by evaluating the Picard group. Based on these two approaches, we also have found braiding statistics between electric and magnetic charges.

In this work, we have considered Abelian topological phases on connected graphs. It would be interesting to extend our study to non-Abelian topological phases. Due to the non-Abelian statistics, the fusion rules, described by the Laplacian, would become more complicated, which probably allows us to explore more interesting interplay between fractional excitations and

graphs. Studying fermionic analogue of our model would be an another intriguing direction. To this end, one has to introduce directed graphs, which incorporates the direction of an edge between two vertices. Algebraic study on such graphs could lead us to new fermionic topological phases. It would be also interesting and important to address whether one can establish an effective field theory description of our model to see any universal data, such as self-statistics, is fully captured by the Laplacian, analogously to the $K$-matrix description of topologically ordered phases [27]. Such investigation would contribute to exploring new types of topological field theory.

Last but not least, it is important to ask whether our model is useful for practical purposes, such as quantum error corrections. To this end, one needs to study the stability of the closed loops of electric and magnetic charges and investigate which graph can host stable loops. By setting the length of 1D line in the $y$-direction, $n_y$ to be large so that closed loops in the $y$-direction is stable against local perturbations, one can concentrate on studying stability of closed loops in the $x$-direction. As seen from (50) (57) and in the case of any connected graph, there is always a closed loop which has the form $\mathbf{r} = k(1, 1, \cdots, 1)^T$ $(k \in \mathbb{Z}_N)$. It is the most stable loop as it consists of operators at every vertex. For other loops, the stability crucially depends on the matrix $P$ given in (2). The stability condition can be studied by evaluating invariant factors of the sub-matrix of the Laplacian. It would be interesting to see whether such condition is associated with other quantities of the graph such as connectivity. Furthermore, one has to study asymptotic behavior of the loops by taking large $n$ limit.

We hopefully come back to these issues for future works.

*Notes added.* After submitting the paper, we were informed by the authors of [28] that the similar model is considered in their upcoming work. We thank Ho Tat Lam for this notification.

## Acknowledgement

We thank Bishwarup Ash for discussion. The research at Weizmann was partially supported by the Koshland Fellowship.

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
