# Peer review of "Anisotropic higher rank $\mathbb{Z}_N$ topological phases on graphs"

_SciPost Physics, doi:SciPost Phys. 14, 106 (2023)_

## Round 1 · Referee Report · Anonymous · 2022-10-28

Strengths
1. The authors introduce a new model and demonstrate that it has interesting and novel characteristics; namely an interplay between the graph structure of the lattice, the ground state degeneracy, and the behaviour of excitations.
2. The derivation and explanation of results are illustrated with simple examples.
3. The layout of the paper is generally clear and easy to follow.
Weaknesses
1. The figures would be much more effective if they were placed on the page where the corresponding concept in the text is first introduced.
2. I found some of the notation to be unwieldy.
Report
In this paper, the authors define a qudit lattice model where the lattice is obtained by vertically stacking multiple copies of an arbitrary graph. They show that this model exhibits dependence of the ground state degeneracy and excitations on the geometry of the graph and the qudit dimension. Results are derived using tools from graph theory, especially the graph Laplacian.
I enjoyed reading the paper and I think that the result is interesting and deserves to be published. The authors go beyond the typical square lattice geometry and show that this change has nontrivial effects. Their work points to new research directions and they detail some of these in their conclusion.
I have some comments that I encourage the authors to address, mostly geared towards making the paper easier to read. See the requested changes section for details.
Requested changes
1. Move the following figures to be on the same page as the corresponding description/equations, I think it would increase their effectiveness.
- Fig. 1 should be on the same page as Eq. 4
- Fig. 2 should be on the same page as Eq. 6
- Fig. 3 should be on the same page as the second paragraph from page 15
2. In Fig. 1a, red squares represent vertices where qudits are placed, but in Fig. 1b, red squares seem to represent copies of $X_1^\dagger$. This is confusing and slowed my understanding of the model. In particular, it was unclear whether there are two vertices in Fig. 1b or two copies of $X_1^\dagger$. It would be good to find some way to differentiate between vertices and operators (maybe circles instead of squares for Fig. 1a?). In Fig. 1b, I think it would also be clearer to write $(X_1^\dagger)^2$ at the central vertex, rather than showing two copies of $X_1^\dagger$. The same applies to double copies of $Z_2$.
3. Section III.E: generalizing the Hamiltonian to a product of two nontrivial graphs may be straightforward, but the lattice formed by such a product is less intuitive. Please include a figure with an example of the product of two graphs.
4. Consider using more descriptive subscripts for the operators currently labeled $X_1(Z_1)$ and $X_2(Z_2)$. For example, $X_v$ and $X_e$ to denote operators acting on vertices and edges, respectively. The current notation also clashes with the string operators $W_{e,x,(1)}$, $W_{e,x,(2)}$ as the subscripts 1 and 2 denote even and odd vertices rather than vertices versus edges. I realize that changing one aspect of the notation may create more trouble than it solves, so this is a soft request.
5. Check for spelling and grammar, I noticed several mistakes. Some examples are "inevitable" (invertible) matrices, "braising" (braiding), and omission of "the", e.g. "we introduce Hamiltonian".
Author: Hiromi Ebisu on 2022-12-18 [id 3144]
(in reply to Report 1 on 2022-10-28)
We thank the referee for careful reading of our manuscript. We are pleased by the fact that he/she enjoys reading our work and his/her support for publication. Below we reply to your comments/suggestions .
-
Following the referee's suggestion, we moved our figures. However, as for figure.2, we were unable to move it to the place you suggested in our tex format (probably due to the size of the figure). We will consult the editor on this issue of the layout in the final procedure of the proof.
-
We have amended our figure 1a so that the points where the clock states (generalized qubit states) are placed are represented by dots, not squares. Also, Notations of (X_1^{\dagger})^2 as well as Z_2^2 have been modified in fig1b.
-
We have added a new figure for an example of the generalized 2D lattice consisting of two connected graphs. (Fig. 5)
-
We would like to stay with the current notations of the operators e.g., X_1, X_2 as the notation may create more trouble than it solves as you said. However, for more transparency of the notations, above eq. 3, we have emphasized that the subscripts are introduced to distinguish the operators that act on the clock states at vertices and the ones at vertical edges in the amended manuscript.
-
We have made wording corrections in the revised manuscript.
Authors
Author: Hiromi Ebisu on 2022-12-18 [id 3145]
(in reply to Report 2 on 2022-12-17)We thank the referee for his/her careful reading of our paper and his/her support for publication.
"Could the authors explain why they call the lattices they consider 2D?
Since they are formed by the product of a 1D line with a graph, do they envision the graph to be laid out along 1D in some way?
Are the number of points in the graph expected to scale with the number of points along the 1D line when taking a thermodynamic limit?"
Yes, it is 2D. We regard the graph as 1D since it consists of 0- and 1-simplices (which correspond to vertices and edges ). Therefore, the lattice composed of the graph and 1D line becomes 2D. For clarity of this point, we have added a footnote [21] in the revised manuscript.
The 2D lattice is constructed in such a way that copies of the graphs are stacked along the 1D line. For instance, copies of the cycle graphs C_3 (triangle) are stacked along the vertical line, giving the lattice portrayed in Fig. 1(a). By construction, the number of points of the graph scales with the number of points of the 1D line in the thermodynamic limit.
"Do the authors know if the models they define satisfy the topological quantum order condition? I.e. there should be no local logical operators acting on the degenerate groundspace, where local is defined with respect to the graph connectivity."
Depending on the graph, there is a logical operator which is local in the sense that it has support of a few neighboring vertices. So far we do not know the condition of no local logical operators. We only can judge whether such a logical operator exists or not by evaluating the kernel of the sub-matrix of the Laplacian on a case-by-case basis. Hopefully we will leave this issue for future studies.

---

## Round 2 · List of Changes

Layout of the figures has been changed. Minor modification has been made in Fig. 1.
For clearer illustration on the generic lattice that we introduce in Sec. III E, a new figure has been added (fig. 5).
A footnote [21] has been added to address why we regard our lattice as 2D. (i.e., why we regard the graph as 1D)

---

## Editorial Decision

published